# Effects of Dietary Lysine Levels on Growth Performance, Nutrient Digestibility, Serum Metabolites, and Meat Quality of Baqing Pigs

**DOI:** 10.3390/ani12151884

**Published:** 2022-07-23

**Authors:** Xuecai Hu, Bin Huo, Jiameng Yang, Kun Wang, Lingjie Huang, Lianqiang Che, Bin Feng, Yan Lin, Shengyu Xu, Yong Zhuo, Caimei Wu, De Wu, Zhengfeng Fang

**Affiliations:** 1Key Laboratory for Animal Disease Resistance Nutrition of the Ministry of Education, Animal Nutrition Institute, Sichuan Agricultural University, Chengdu 611130, China; hxc15181789484@163.com (X.H.); sichuanhuobin@163.com (B.H.); yjm5161@outlook.com (J.Y.); wangkk002714@163.com (K.W.); 71424@sicau.edu.cn (L.H.); che.lianqiang@sicau.edu.cn (L.C.); fengbin@sicau.edu.cn (B.F.); linyan936@163.com (Y.L.); shengyuxu@sicau.edu.cn (S.X.); zhuoyong@sicau.edu.cn (Y.Z.); zhuomuniao278@163.com (C.W.); wude@sicau.edu.cn (D.W.); 2Key Laboratory for Food Science and Human Health, College of Food Science, Sichuan Agricultural University, Ya’an 625014, China

**Keywords:** Baqing pigs, lysine, growth performance, nutrient digestibility, serum metabolites

## Abstract

**Simple Summary:**

This study investigated the effects of different standardized ileal digestible (SID) Lys levels on growth performance, nutrient digestibility, serum metabolites, and carcass and meat traits of Baqing pigs. The results revealed that the addition of dietary SID Lys was beneficial for the average daily feed intake (ADFI), gain to feed ratio (G/F), and average daily gain (ADG) of Baqing pigs at the growth phase. The serum concentrations of triglycerides, lysine, and histidine increased by increasing dietary Lys levels. Compared with the treatment three group, dietary SID Lys addition content at treatment four increased the shear force of the longissimus dorsi muscle, but it did not affect the other carcass and meat traits. The SID Lys requirement of 20–40 kg, 40–60 kg, and 60–90 kg of Baqing pigs fed corn–soybean meal-based diets is estimated to be 0.92%, 0.66%, and 0.55% of the diets, respectively.

**Abstract:**

This study was carried out to determine the Lys requirements of Baqing pigs and the effects of different dietary lysine levels on growth performance, apparent nutrient digestibility, serum metabolites, and carcass and meat traits. A total of 120 castrated Baqing pigs were selected by body weight and randomly assigned to five dietary treatments with six replicate pens (4 pigs per pen, castrated) per treatment in a randomized complete block design. Five diets in mash form were formulated to contain SID Lys at 0.56%, 0.68%, 0.80%, 0.92%, and 1.04% of diet in phase 1 (20–40 kg), at 0.45%, 0.54%, 0.63%, 0.72%, and 0.81% of diet in phase 2 (40–60 kg), and at 0.39%, 0.47%, 0.55%, 0.63%, and 0.71% of diet in phase 3 (60–90 kg), respectively. The results showed that the bodyweight of pigs was not affected by dietary SID Lys content during each period. However, the addition of dietary SID Lys linearly reduced F/G in the first period and quadratically increased ADG during the second period (*p* < 0.05). The digestible energy (DE) was increased linearly and quadratically in the first phases with the dietary increased SID Lys levels, while DE was reduced in the third and second phases (*p* < 0.05). Increasing SID Lys contents linearly increased the serum TG concentration and quadratically decreased the serum GLU concentration, while linearly reducing the serum HDLC concentration of first period pigs (*p* < 0.05). Serum concentrations of TP, TG, TC, and LDLC were increased linearly with the increasing dietary SID Lys levels in the second period (*p* < 0.05). The serum concentrations of Lys increased quadratically, and histidine increased linearly with the increased dietary SID Lys levels (*p* < 0.05). Compared with the treatment three group, dietary SID Lys addition content at treatment four increased the shear force of the longissimus dorsi muscle (*p* < 0.05), but it did not affect the other carcass and meat traits. The optimal SID Lys requirement of 20–40 kg, 40–60 kg, and 60–90 kg of Baqing pigs fed corn–soybean meal-based diets is estimated to be 0.92%, 0.66%, and 0.55% of the diets by the quadratic curve models, respectively.

## 1. Introduction

Chinese native pig breeds are a crucial component of the world’s pig genetic resources and an essential source of Chinese premium quality pork production [1]. The Baqing pig is an excellent indigenous pig breed—now widely bred in Southwest China—due to its outstanding advantages of indigenous robustness, roughage-resistance, exceptional prolificacy, and desirable meat quality [2]. However, compared with the high-performing pigs, the Baqing pig has a much lower lean percentage and relatively slower growth rate, which means a lower production efficiency that limits its use for commercial production [3,4]. Currently, diets for the Baqing pig are mainly formulated based on the nutrition recommendation tables of the National Research Council (NRC, 2012) owing to a dearth of information about proper nutrient requirements of indigenous pigs [5]. Obviously, the same diet is not appropriate due to the different characteristics of this indigenous breed [6]. Therefore, it is necessary to find the optimal nutritional requirements of the Baqing pig for the improvement of its production performance and the promotion of indigenous pig breed resource development.

Dietary amino acids (AA) are used mainly for organism protein synthesis, and the AA concentration in the diet is directly related to the performance of pigs [7,8]. Lysine (Lys) is the first limiting AA in swine fed corn–soybean meal diets and is also generally considered as a reference AA for the relative amounts required for the other AA [5]. Currently, over 95% of total Lys produced is used as a feed additive to improve the utilization rate of crude protein in the world. Andretta et al. [9] stated that dietary Lys supplementation can maintain the dietary amino acid balance, improve feed conversion rate, and indirectly reduce nitrogen excretion. Wu [10] stated that Lys is absorbed directly by the intestine for protein synthesis and other metabolic processes, such as the regulation of nitric oxide synthesis, mineral accumulation, antiviral activity, protein methylation, and O-linked glycosylation. Meanwhile, Lys also provides structural components for the synthesis of carnitine, which participates in fat metabolism and regulates metabolic balance in animals [11]. Furthermore, Lys can act as a regulatory factor for the synthesis and secretion of growth hormone, insulin, and insulin-like growth factor-1, thereby promoting the release of endocrine hormones [12]. In growing pigs, deficiency or excess of dietary Lys cannot only reduce or increase the abundance of its transporters in vivo but also decrease or raise nitrogen retention and protein turnover in the organism, thereby affecting its production performance, nutrient digestibility, serum metabolic parameters, and carcass characteristics [13,14]. Therefore, the Lys requirement of pigs has always been an important research topic in nutrition, and it is the basis for establishing an ideal protein model for pigs.

Therefore, the objective of the current study was to investigate the effects of dietary standardized ileal digestible (SID) Lys levels on growth performance, nutrient digestibility, serum metabolites, and carcass and meat traits of Baqing pigs, and to provide a preliminary estimate on the Lys requirement in the stage of growing and fattening to provide scientific and objective data as a reference for the optimal nutritional strategies of Chinese indigenous pig breeds.

## 2. Materials and Methods

The experiment was conducted in accordance with the Chinese Guidelines for Animal Welfare and the experimental procedures were approved by the Animal Care and Use Committee of the Animal Nutrition Institute, Sichuan Agricultural University (Ethics Approval Code: SCAUAC201806-9).

### 2.1. Dietary Treatments, Animals and Housing

The study was carried out at the Teaching and Research Base of the Sichuan Agricultural University, Animal Nutrition Institute (Ya’an, China). The experimental period included three phases based on the body weight: namely, phase 1 (20–40 kg), phase 2 (40–60 kg), and phase 3 (60–90 kg). A total of 120 castrated Baqing pigs at approximately 70 days of age with an average initial body weight of 18.65 ± 8.85 kg were allotted randomly to five dietary treatments based on sex and body weight. Each treatment had six replicates with four pigs (two barrows and two gilts) per pen. In each phase, five experimental diets were prepared by supplementation of L-lysine HCl (Yipin Biotechnology Co., Ltd., Ningxia, China; with purity ≥ 98.5%) to the basal diet. Dietary SID Lys levels were designed in T1 to T5 to 70%, 85%, 100%, 115%, and 130% of the recommended levels of the China Nutrition Requirement of Swine (NY/T65-2004), respectively (Table 1) [15]. In each phase, a basal diet, with varying levels of SID Lys but adequate in terms other nutrients, was formulated according to the nutrition requirement of local-type pigs as recommended by the China Nutrition Requirement of Swine (NY/T65-2004) (Table 2). The analyzed composition of diets is shown in Appendix A. All pigs were housed in conventional facilities with a half-slatted concrete floor and had free access to feed and water. The animal house was regularly disinfected and kept the ambient temperature and ventilation in commercial conditions.

### 2.2. Growth Performance

During the experimental period, pigs were weighed individually at the end of each phase after overnight fasting. Feed intake of each pen was recorded weekly to determine average daily gain (ADG), average daily feed intake (ADFI), and feed to gain ratio (F/G). Feed samples (1 kg) were collected from each diet phase and then stored at −20 °C until further analysis [16]. At the end of each phase, pigs fasted for 12 h (free access to water) before collecting blood samples. A total of 10 mL of blood from each pig was collected by jugular venipuncture into non-heparinized tubes. Subsequently, samples were centrifuged (3500× *g*, 4 °C, 15 min) to separate the serum and stored at −20 °C until analysis [16]. At the end of each phase, five diets containing 0.3% chromic oxide (Cr_2_O_3_) as a digestibility marker were offered for five days as an adaption period, and then this diet was continuously fed for three days. Fecal samples (100 g per pig) were collected for three consecutive days from the rectum after pigs had free access to their diet for 30 min. All fecal samples were fixed with 10% dilute hydrochloric acid and toluene, then stored at −20 °C prior to analysis [17,18].

### 2.3. Carcass and Meat Traits Measurement

At the end of the experiment, twelve pigs (six in each treatment) were selected from treatment three and treatment four (the optimum Lys level showed the highest growth performance) to determine carcass and meat traits (left side of each carcass). Pigs fasted for 24 h (free access to water) and then were slaughtered; segmentation was according to Technical Procedures for Determination of Performance of Lean Breeding Pigs (GB 8467-87) [19]. Carcass traits were measured according to the Technical Regulation for Testing of Carcass Traits in Lean-Type Pigs (NY/T825-2004) [20]. Meat traits determined reference to the Determination of Livestock and Poultry Meat Quality (NY/T133-2007) [21]. The pH of the longissimus was measured with a pH meter (pH-start, Matthaus, Germany) 45 min after slaughter and then stored at 4 °C and measured again after 24 h. Meat color (lightness L*, redness a*, and yellowness b*) were tested with a Minolta Chromameter (CR 400, Konica Minolta Inc., Tokyo, Japan). Shear force was determined using a Stable Micro Systems Texture Analyzer (TA-XT Plus, Stable Micro Systems, Surrey, UK). Intramuscular fat was determined using the Soxhlet extraction method.

### 2.4. Chemical Analysis

The fecal and feed samples were dried in a fan-forced oven at 60 °C for 72 h and then ground to pass a 0.45 mm sieve W.S. Tyler mill for chemical analysis. Proximate analysis of the experimental diets and feces samples were performed using standard Association of Official Analytical Chemists methods [18]. The samples were dried at 105 °C in a drying oven to a constant weight to determine dry matter (DM) content. Crude protein (CP) was measured by Auto Kjeldahl Analysis Equipment (Kjeltectm 8400, FOSS, Hilleroed, Denmark). Gross energy (GE) was measured using an automatic adiabatic oxygen bomb calorimeter (Parr 6400 Calorimeter; Moline, IL, USA), and crude ash was combusted at 550 °C in an SX2-12-10 muffle furnace (Rongfeng Corporation, Shanghai, China). The Cr content in the ashed samples of the diets and feces was analyzed spectrophotometrically (ContrAA^®^ 700, Analytik Jena AG, Jena, Germany) after acid digestion. Apparent nutrient digestibility was calculated using the Cr_2_O_3_ content in feed and fecal samples based on the chemical analysis results [13,17,18]. The 350 μL serum samples were diluted with 10% sodium sulfosalicylic at 1:2, then centrifuged (12000× *g*, 4 °C 10 min) after standing for 5 min, and the clear supernatant was filtered through a 0.45 m filtration membrane after centrifuging. Subsequently, the amino acids in the serum were analyzed by using an Amino Acid Analyzer (L-8800, Hitachi Ltd. Tokyo, Japan) [13]. The 500-μL serum samples were centrifuged (3000× *g*, 4 °C, 3 min), and, then, the serum metabolites were determined by an automatic biochemical analyzer (7020, Hitachi Ltd., Tokyo, Japan) [16].

### 2.5. Statistical Analysis

Data on growth performance, apparent digestibility, serum metabolites, and amino acid concentration were statistically analyzed by a generalized linear model (GLM) using a normal distribution with SAS, version 9.4 (SAS Institute, Inc., Cary, NC, USA, 2014) [22]. Data of carcass characteristics and meat traits were analyzed by paired sample t-test according to the weight of the swine. Each pen was considered the experimental unit for growth performance and the individual pig as the experimental unit for other indices. According to the indices of the ADG or F/G, which were consistent with a quadratic curve, the quadratic regression equation Y = aX^2^ + bX + c (Y: index, X: dietary SID Lys concentration) related to the SID Lys requirements was established using the curve regression in regression analysis by the procedure of SAS. Results are presented as least squares means and standard error of means (SEM). The significance of the model was set at *p* < 0.05.

## 3. Results

### 3.1. Growth Performance and Estimation of Lysine Requirements

The effects of dietary SID Lys levels on the growth performance of Baqing pigs are presented in Table 3. During the first period, the ADLI increased linearly (*p* < 0.001) and tended to increase quadratically (*p* = 0.096) as dietary SID Lys concentration increased, while F/G reduced linearly (*p* = 0.039). During the second period, the ADG increased quadratically (*p* = 0.010), and ADLI increased linearly (*p* < 0.001) with increasing dietary Lys levels. In the third phase, the ADLI increased linearly (*p* < 0.001) and quadratically (*p* = 0.025) with the increasing Lys levels. Also, the ADFI tended to reduce quadratically as dietary Lys concentration increased in each period. The body weight at the end of each feeding period was not different among treatments, but it is worth noting that the body weight and ADG in each growth period had the greatest numerical value in the T4 group.

As shown in Figure 1 and Figure 2, the quadratic curve models estimated the minimum F: G at 0.92% SID Lys level of Baqing pigs for 20–40 kg. The curvilinear plateau models described the requirement for the maximum performance at 0.66% SID Lys level for ADG of Baqing pigs for 40–60 kg. The dietary SID Lys requirement of Baqing pigs at 60–90 kg was 0.55% based on the recommended level of the China Nutrition Requirement of Swine (NY/T65-2004) due to the dietary Lys levels not having affected the performance of 60–90 kg pigs [15].

### 3.2. Apparent Fecal Energy and Nutrient Digestibility

The effects of dietary SID Lys levels on the apparent fecal energy and nutrient digestibility of Baqing pigs are presented in Table 4. In the first phase, the CA increased linearly (*p* = 0.018), and the DE increased linearly and quadratically (*p* < 0.001) with increasing dietary Lys levels. In the second period, the DE reduced linearly and quadratically (*p* < 0.001), while the CA reduced quadratically (*p* = 0.025) with an increasing level of dietary Lys. In the third phase, the DE reduced linearly and quadratically with the increase in dietary Lys levels (*p* < 0.001). However, the digestibility of DM and CP were not significantly affected by dietary treatment.

### 3.3. Serum Metabolites

The effects of dietary SID Lys levels on serum metabolites of Baqing pigs are shown in Table 5. In the first period, the increasing dietary Lys level linearly increased the serum TG concentration (*p* < 0.001) and quadratically reduced the serum GLU concentration (*p* = 0.017) while linearly reducing the serum HDLC concentration (*p* = 0.019). However, the dietary Lys levels did not affect serum concentrations of UREA, TC, and LDLC. In the second period, the serum concentrations of TP, TG, TC, and LDLC increased linearly with increasing dietary Lys levels (*p* < 0.05). However, there were no significant differences in the serum concentration of UREA, HDLC, and GLU among diet treatments. In the third phase, the serum metabolite parameters were not significantly affected by dietary treatments.

### 3.4. Serum Amino Acid Concentration

The effects of dietary SID Lys levels on serum amino acid concentration of Baqing pigs are shown in Table 6. The serum concentrations of Lys and histidine increased quadratically (*p* = 0.010) and linearly (*p* = 0.002), respectively, with an increasing level of dietary Lys. The serum alanine concentration in the T3 group was significantly higher than in other groups (*p* = 0.012). Serum concentrations of serine, leucine, and phenylalanine tended to change (0.05 < *p* < 0.1) as dietary Lys levels increased. However, dietary Lys levels did not affect the other serum amino acid concentrations.

### 3.5. Carcass and Meat Traits

The effects of dietary SID Lys levels on the carcass and meat traits of Baqing pigs are presented in Table 7 and Table 8. The carcass characteristics of pigs were not significantly affected by the dietary SID Lys level (*p* > 0.05). The addition of dietary SID Lys increased the shear force of the longissimus dorsi muscle (*p* = 0.007), whereas it did not affect the other meat traits (*p* > 0.05).

## 4. Discussion

Lysine acts as an essential AA for protein synthesis and plays a role in pig growth performance and productivity [23]. The present study aimed to determine the response of pigs from the Chinese indigenous genotype to variables in the SID Lys level from 20 to 90 kg bodyweight and therefore estimate the optimal dietary SID Lys requirements of Baqing pigs at growth-finishing periods. Previous studies have clarified that Lys supplementation in the basal diet improved the growth performance and nutrient digestibility of growing and finishing pigs [24,25]. Similarly, Oresanya et al. [26] illustrated that the BW and ADG of weaned pigs increased quadratically with increasing dietary Lys concentration. Corino et al. [27] reported that both ADG and G/F of pigs increased in response to the increased dietary Lys levels. Rodriguez-Sanchez et al. [28] found a linear increase in ADG and ADFI as total dietary Lys was increased from 0.6% to 0.7% of diet from 100 to 128 kg body weight. The current study stated that the F/G linearly reduced in the first period and the ADG quadratically increased during the second period as the dietary SID Lys inclusion rate increased but was not affected in the finishing period. Tous et al. [29] observed that pigs needed to eat more to achieve the BW of the control group, although the reduction in weight gain did not reach significance in the pigs fed the low Lys diet. Also found was an increase in G/F without significant effects on ADFI or ADG when dietary Lys was reduced and the protein kept at the same level as the control diet [29]. It has been several years since the response of Chinese indigenous pigs to Lys was reviewed, and their Lys requirements likely changed with differences in pig genotypes and growth periods. Therefore, the results of dietary Lys levels were given in a wide range. Chang [30] showed that the optimum requirement for apparent ileal digestible Lys in 30–60 kg and 60–90 kg black pigs in southern Henan province of China were 0.64% and 0.84% of the diet, respectively. Yang [31] observed that the total Lys requirements of the Chinese southwest pigs (Landrace × Rongchang) were 0.71% and 0.64% of the diet for 20–50 kg and 50–80 kg, respectively. Yuan et al. [32] suggested that the digestible Lys requirement of Yantai black pig (15–30 kg) was 0.85% of the diet. In the present study, the SID Lys requirement for minimum F/G and maximum daily gain between 20–40 kg and 40–60 kg BW of Baqing pigs ranged from 0.92% and 0.66%, respectively. However, the estimated values of optimal SID Lys in this study were slightly lower than the ideal SID values of 25 to 50 and 50 to 75 kg pigs by the NRC [5]. Normally, large-sized or high lean breed pigs have higher Lys requirements than small-physique or fat-type pig species at the same growth phase [6]. In addition, when the Lys requirement was measured by concentration in dietary ratio, the Lys requirement of pigs decreased with the increase of age and body weight, which was due to the increase in feed intake and fat deposition rate.

It is well known that an appropriate Lys level in diet could promote the development of animal digestive organs and improve the digestion and utilization of feed nutrients. Zeng et al. [13] reported that an appropriate density of 0.95% dietary Lys makes the dietary nutrient apparent digestibility reach a better level in 20 kg pigs. While reducing the total Lys content to 0.65% significantly reduces the apparent digestibility of crude protein, energy, dry matter, and phosphorus. He et al. [33] indicated that dietary supplementation with Lys enhanced villus height, crypt depth, and expression of cationic amino acid transporters genes in the jejunum. Jie et al. [34] found that the dietary Lys restriction inhibited intestinal Lys transport and promoted feed intake, which might be associated with the gut microbiome. Wu et al. [35] clarified that functional AAs (arginine, glutamine, leucine, etc.) serve essential regulatory functions in nutrient metabolism, protein turnover, and immune function and further enhance the efficiency of feed utilization by pigs. Although Lys is not considered a functional AA, it may affect nutrient absorption and metabolism by adjusting the absorption of some functional AAs or by regulating endocrine hormone release [36]. Yang [37] verified that the apparent total tract digestibility of dry matter, gross energy, crude protein, and minerals were increased linearly in Yacha pigs of 50 to 90 kg with the increase of dietary SID Lys level from 0.30% to 0.66%. Chang [30] found that increasing the dietary digestible Lys level from 0.64% to 0.84% in the 60–90 kg stage of Yunan black pigs can significantly improve the apparent digestibility of phosphorus and crude fat. In the present study, the digestibility of crude ash and digestible energy in Baqing pigs at the 20–40 kg stage increased linearly with the increase in dietary SID Lys levels, and was highest at the 0.92% dietary SID Lys level group. At the 40–60 kg stage, the digestible energy reduced linearly and quadratically with increasing dietary SID Lys levels, and the best digestible energy was obtained by pigs fed the 0.72% SID Lys diet. However, the higher dietary Lys level may cause an AA imbalance. In addition, Lys is antagonistic to some AA, such as phenylalanine, cysteine, arginine, and methionine, which may be affecting the absorption of AA and nutrients [36]. Herein, the dietary Lys level of the T5 group may be excessive for Baqing pigs. These may be the reasons for the lower nutrient digestibility in the T5 group. Therefore, these studies have shown that an appropriate level of dietary Lys can increase the apparent nutrient digestibility of pigs and reduce the excretion of nutrients in feces, thereby further improving the efficiency of pigs’ utilization of nutrients.

Changes in serum biochemical indicators reflect an alteration in the permeability of tissue cells and variation in the organism’s metabolic functions [38]. Malawi et al. [39] suggested that the changes in dietary nutrient composition would affect the distribution of serum metabolites, and its mechanism might be regulating the transcription of enzymes related to nitrogen and lipid metabolism. Serum lipid content is an important indicator reflecting lipid metabolism in animals. The content of triglycerides and total cholesterol in the blood can reflect the lipid metabolism in animals [40]. This study showed that the serum content of triglycerides and total cholesterol increased linearly with the increase in dietary Lys levels in the second phase, both of which were the highest in T4 group. Mule et al. [41] reported that serum cholesterol in growing and finishing pigs was correlated negatively with Lys intake. Zou et al. [42] and Zeng et al. [43] stated that dietary Lys levels did not significantly affect piglets’ serum total cholesterol content. Zeng et al. [13] exhibited that dietary Lys levels had no significant effect on the serum triglyceride concentration of growing pigs. These inconsistent results in different research may be associated with pig breeds, test cycles, growth stages, and Lys addition gradients. The triglycerides of pigs are synthesized primarily in adipose tissue, and the continuous accumulation of triglycerides will lead to the deposition of fat cells. The increase of triglycerides is the result of fat synthesis greater than lipolysis [42]. The Baqing pig is a fat-type pig breed and has a strong ability to deposit fat, which may be the main reason for the inconsistency between the results of this study and those of previous studies. The serum total protein content is an indicator that reflects protein metabolism in the organism, and the higher total protein concentration denotes a higher level of protein deposition in the organism. In this study, the serum total protein level in pigs increased linearly with the increase of dietary SID Lys levels at the 40-60 kg stage. Similarly, Regmi et al. [44] reported that plasma albumin concentration increased with increasing dietary total Lys level (0.43%, 0.71%, 0.98%). Thus, the results of this study stated that protein deposition could increase by adding Lys hydrochloride at a low protein level, which could improve the production performance of Baqing pigs.

The serum amino acid content is usually affected by the amino acid content in the feed. The concentration and ratio of various free amino acids in animal serum can reflect the composition of dietary amino acids and the metabolism of amino acids in the body, which often are important indicators for evaluating amino acid metabolism [10]. The current study stated that the serum concentrations of Lys increased quadratically and histidine increased linearly by increasing dietary SID Lys levels. This is in agreement with Zhou et al. [45], who indicated that serum free Lys, alanine, and histidine levels increased linearly as dietary SID Lys levels (0.67%, 0.72%, 0.79%, 0.86%) increased. Zeng et al. [13] found that the serum Lys concentrations increased with increased dietary Lys levels, whereas the serum concentrations of isoleucine, histidine, phenylalanine, threonine, and valine decreased. Regmi et al. [46] stated that dietary Lys deficiency reduced the expression of amino acid transporters in the small intestine and plasma concentration of free amino acid, thereby further causing a series of interrelated negative consequences for pig health and production performance. Cho et al. [47] observed that increasing the Lys to DE ratio increased the apparent digestibility of essential amino acids except for leucine, regardless of energy density. Several studies have found that diets with low protein or Lys levels can increase the intramuscular fat content of pork, thereby improving pork quality, e.g., enhancing pork flavor and tenderness [48]. However, it may reduces carcass lean meat percentage and increases fat deposition [48]. Conversely, increasing dietary ideal protein levels significantly reduced intramuscular fat content and marbling scores of pork [49]. However, adding Lys in protein deficient diets could reduce backfat thickness and increase eye muscle area and lean meat rate [50]. Additionally, previous studies stated that dietary Lys at an appropriate levels could improve carcass traits and meat quality by increasing muscle volume and fiber diameter [51]. In the present study, increasing dietary Lys content increased the shear force of the longissimus dorsi muscle, i.e., reduced pork tenderness. However, it did not affect the other carcass and meat traits. Similar results were also reported by Zhang et al. [52] and Yang [37], who evaluated the levels of Lys in growing-finishing pigs. The reason may be that the contents of total protein and other amino acids fed have already met the requirement of pigs. Thus, the dietary SID Lys addition at an appropriate level might promote the feed utilization and performance of Baqing pigs but slightly reduce the tenderness of pork without reducing its other pork quality.

## 5. Conclusions

The optimum dietary SID Lys levels improved growth rate, apparent nutrient digestibility, meat traits, and serum amino acid concentration of Baqing pigs. The daily SID Lys requirement of 20–40 kg, 40–60 kg, and 60–90 kg Baqing pigs was estimated to be 10.35 g/d, 16.37 g/d, and 17.75 g/d, respectively, and the optimal dietary SID Lys levels were 0.92% (0.69 g/MJ ME), 0.66% (0.72 g/MJ ME) and 0.55% (0.64 g/MJ NE) of diet, respectively.

## Figures and Tables

**Figure 1 animals-12-01884-f001:**
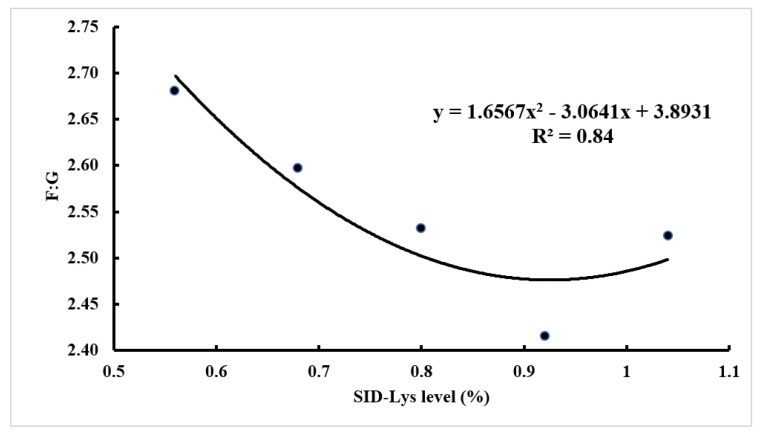
Quadratic regression trends between dietary SID Lys level and F:G of Baqing pigs for 20–40 kg.

**Figure 2 animals-12-01884-f002:**
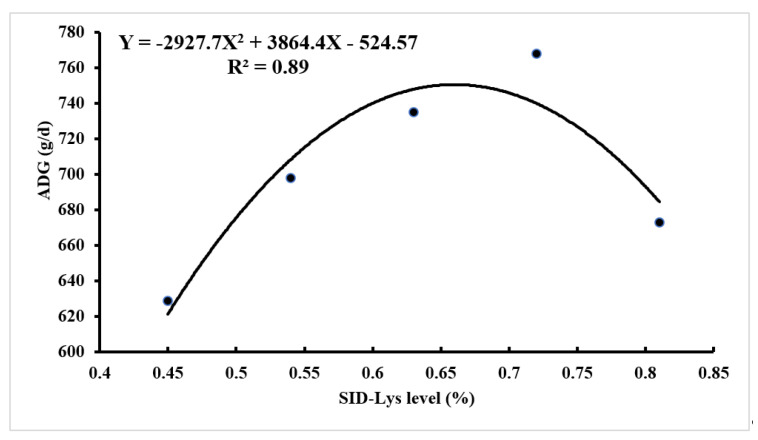
Quadratic regression trends of between dietary SID Lys level and ADG of Baqing pigs for 40–60 kg.

**Table 1 animals-12-01884-t001:** The dietary SID lysine and L-lysine HCl supplementation levels of each treatment in the feeding phases.

Item	T1	T2	T3	T4	T5
SID Lys levels (%)					
20–40 kg (Phase 1)	0.56	0.68	0.80	0.92	1.04
40–60 kg (Phase 2)	0.45	0.54	0.63	0.72	0.81
60–90 kg (Phase 3)	0.39	0.47	0.55	0.63	0.71
Addition of L-Lys HCl levels (%)					
20–40 kg (Phase 1)	0.030	0.183	0.340	0.488	0.640
40–60 kg (Phase 2)	0.018	0.145	0.260	0.375	0.503
60–90 kg (Phase 3)	0.030	0.132	0.235	0.335	0.045

**Table 2 animals-12-01884-t002:** Ingredients of basal diets.

Item	Body Weight		
	20–40 kg	40–60 kg	60–90 kg
Ingredients, %			
Corn, CP 7.8%	67.36	64.84	60.86
Soybean meal, CP 44.2%	14.60	8.70	5.25
Wheat bran, CP 15.7%	7.37	10.22	15.16
Soybean hull, CP 10.27%	4.05	9.00	10.55
Corn gluten meal, CP 60%	1.80	2.80	3.80
Soybean oil	2.05	2.05	2.05
Limestone	1.10	1.00	1.20
Dicalcium phosphate	0.65	0.45	0.20
Sodium chloride	0.35	0.35	0.35
Chloride choline	0.12	0.12	0.12
Sodium sulphate	0.10	0.10	0.10
Phytase, 10,000 U/g	0.01	0.01	0.01
NSP enzyme	0.01	0.01	0.01
Threonine	0.12	0.055	0.04
Tryptophan	0.03	0.02	0.02
Vitamin premix ^(1)^	0.03	0.03	0.03
Mineral premix ^(2)^	0.15	0.15	0.15
Antioxidant	0.03	0.03	0.03
Fungicide	0.07	0.07	0.07
Total	100.0	100.0	100.0
Composition, % ^(3)^			
DE, kcal/kg	3386	3288	3215
ME, kcal/kg	3282	3194	3123
NE, kcal/kg	2480	2405	2345
CP, %	14.54	13.27	12.91
SID AA, %			
SID Lysine	0.56	0.44	0.39
SID Methionine	0.22	0.20	0.20
SID Threonine	0.55	0.43	0.39
SID Tryptophan	0.16	0.13	0.12
SID Isoleucine	0.48	0.42	0.39
SID Valine	0.54	0.49	0.47
Calcium, %	0.65	0.57	0.58
Total phosphorus, %	0.48	0.43	0.41
Available phosphorus, %	0.25	0.22	0.20

^(1)^ Provided the following per kg of diet: vitamin A, 3000–9600 IU; vitamin D_3_, 600–6000 IU; vitamin E, 30 mg; vitamin K, 2.4–6 mg; vitamin B_1_, 0.6 mg; vitamin B_2_, 3 mg; vitamin B_6_, 1.2 mg; Nicotinamide, 18 mg. ^(2)^ Provided the following per kg of diet: Zn, 27–55 mg; Mn, 14–30 mg; Fe, 45–96 mg; Cu, 10 mg; I, 0.2 mg; Se, 0.3 mg. ^(3)^ Nutrient levels were calculated values.

**Table 3 animals-12-01884-t003:** Effect of dietary SID lysine levels on growth performance of Baqing pigs at 20–90 kg.

Item	Treatment					SEM	*p*-Value		
	T1	T2	T3	T4	T5		GLM	Linear	Quadratic
Phase 1 (20–40 kg)									
Initial body weight, kg	19.38	19.35	19.40	19.23	19.19	1.075	0.996	0.722	0.880
BW ^1^, kg	41.21	41.26	40.58	43.06	42.15	2.417	0.446	0.247	0.741
ADG ^2^, kg/d	0.55	0.55	0.53	0.60	0.57	0.048	0.172	0.115	0.644
ADFI ^3^, kg/d	1.46	1.41	1.38	1.44	1.43	0.093	0.200	0.940	0.059
F/G ^4^	2.68	2.60	2.53	2.42	2.52	0.177	0.150	0.039	0.227
ADLI ^5^, g/d	8.17 ^e^	9.53 ^d^	10.69 ^c^	13.21 ^b^	14.94 ^a^	0.762	<0.001	<0.001	0.096
Phase 2 (40–60 kg)									
Initial body weight, kg	41.27	41.26	40.58	43.06	42.15	2.417	0.466	0.247	2.741
BW, kg	63.92	64.80	63.97	67.40	61.13	4.642	0.978	0.789	0.668
ADG, kg/d	0.71 ^a^	0.74 ^a^	0.73 ^a^	0.76 ^a^	0.59 ^b^	0.079	0.046	0.133	0.010
ADFI, kg/d	1.96	2.09	2.11	2.27	2.07	0.183	0.118	0.131	0.093
F/G	2.76	2.82	2.89	2.97	3.51	0.320	0.394	0.779	0.055
ADLI, g/d	8.99 ^d^	10.65 ^c^	13.33 ^b^	16.13 ^a^	16.65 ^a^	1.117	<0.001	<0.001	0.287
Phase 3 (60–90 kg)									
Initial body weight, kg	63.92	64.80	63.97	67.40	61.13	4.642	0.978	0.789	0.668
BW, kg	87.20	87.29	88.15	91.61	84.32	5.753	0.462	0.906	0.273
ADG, kg/d	0.63	0.61	0.65	0.65	0.63	0.061	0.697	0.868	0.646
ADFI, kg/d	2.62	2.67	2.68	2.82	2.41	0.264	0.193	0.478	0.072
F/G	4.17	4.41	4.31	4.35	3.83	0.354	0.824	0.728	0.409
ADLI, g/d	9.29 ^d^	11.44 ^c^	15.04 ^b^	16.95 ^a^	17.41 ^a^	1.318	<0.001	<0.001	0.025

BW ^1^: Body weight; ADG ^2^: Average daily gain; ADFI ^3^: Average daily feed intake; F/G ^4^: Feed to gain ratio; ADLI ^5^: Average daily SID Lys intake. ^a, b^^, c^^, d, e^ Means within a row with no common superscripts are significantly different (*p* < 0.05).

**Table 4 animals-12-01884-t004:** Effect of dietary SID lysine levels on apparent nutrient digestibility of Baqing pigs at 20–90 kg.

Item	Treatment					SEM	*p*-Value		
	T1	T2	T3	T4	T5		GLM	Linear	Quadratic
Phase 1 (20–40 kg)									
DM ^1^, %	82.43	82.53	82.09	83.16	82.68	1.493	0.798	0.567	0.875
GE ^2^, %	82.48	82.68	82.00	83.10	82.67	1.483	0.782	0.684	0.821
CP ^3^, %	79.83	80.68	80.15	82.87	80.01	2.561	0.253	0.448	0.297
CA ^4^, %	46.33 ^b^	46.38 ^b^	46.75 ^b^	52.85 ^a^	49.58 ^ab^	3.933	0.033	0.018	0.883
DE ^5^, Mcal/kg	13.63 ^c^	13.69 ^b^	13.40 ^e^	13.77 ^a^	13.62 ^d^	0.000	<0.001	<0.001	<0.001
Phase 2 (40–60 kg)									
DM, %	85.00	83.97	83.44	84.90	83.21	1.522	0.173	0.191	0.781
GE, %	85.06	83.78	83.29	84.74	82.90	1.531	0.099	0.102	0.731
CP, %	83.36	82.22	81.08	82.71	81.68	2.267	0.470	0.337	0.398
CA, %	56.20	45.69	46.34	54.45	54.67	7.906	0.074	0.582	0.025
DE, Mcal/kg	14.09 ^b^	13.90 ^c^	13.89 ^d^	14.13 ^a^	13.85 ^e^	0.000	<0.001	<0.001	<0.001
Phase 3 (60–90 kg)									
DM, %	82.27	81.75	81.20	82.98	82.82	1.251	0.109	0.161	0.124
GE, %	82.67	81.66	81.17	82.96	82.64	1.264	0.129	0.147	0.214
CP, %	81.40	80.80	78.67	80.85	80.99	2.183	0.249	0.7900	0.0950
CA, %	44.75	47.61	44.02	49.00	45.39	5.470	0.497	0.708	0.607
DE, Mcal/kg	13.58 ^a^	13.36 ^d^	13.22 ^e^	13.56 ^b^	13.45 ^c^	0.000	<0.001	<0.001	<0.001

DM ^1^: Dry matter; GE ^2^: Gross energy; CP ^3^: Crude protein; CA ^4^: Crude ash; DE ^5^: Digestible energy. ^a^^, b^^, c^^, d, e^ Means within a row with no common superscripts are significantly different (*p* < 0.05).

**Table 5 animals-12-01884-t005:** Effect of dietary SID lysine levels on serum metabolites of Baqing pigs at 20–90 kg (mmol/L).

Item	Treatment					SEM	*p*-Value		
	T1	T2	T3	T4	T5		GLM	Linear	Quadratic
Phase 1 (20–40 kg)									
UREA	2.25	2.43	1.69	2.23	2.04	0.666	0.383	0.471	0.582
TP ^1^	44.63	41.75	36.80	44.18	35.87	6.169	0.057	0.070	0.878
TG ^2^	0.27 ^b^	0.22 ^b^	0.26 ^b^	0.67 ^a^	0.48 ^a^	0.116	<0.001	<0.001	0.643
TC ^3^	1.63	1.51	1.38	1.57	1.23	0.348	0.322	0.115	0.816
LDLC ^4^	0.44	0.41	0.43	0.42	0.30	0.134	0.362	0.129	0.275
HDLC ^5^	0.40	0.35	0.28	0.32	0.28	0.085	0.098	0.019	0.316
GLU ^6^	3.52 ^a^	2.83 ^ab^	2.43 ^b^	3.31 ^a^	2.87 ^ab^	0.452	0.003	0.18	0.017
Phase 2 (40–60 kg)									
UREA	2.83	3.87	3.01	4.18	3.41	0.965	0.116	0.257	0.294
TP	49.68 ^b^	57.25 ^ab^	51.12 ^b^	64.60 ^a^	66.95 ^a^	10.222	0.022	0.004	0.562
TG	0.57 ^b^	0.64 ^b^	0.91 ^b^	1.32 ^a^	0.88 ^b^	0.295	0.002	0.002	0.060
TC	2.21 ^b^	2.37 ^b^	2.27 ^b^	3.42 ^a^	3.03 ^ab^	0.692	0.018	0.006	0.888
LDLC	0.54	0.54	0.69	0.97	0.79	0.285	0.077	0.020	0.595
HDLC	0.51	0.64	0.48	0.68	0.71	0.203	0.209	0.104	0.577
GLU	3.15	4.16	3.49	4.45	4.48	1.147	0.198	0.057	0.853
Phase 3 (60–90 kg)									
UREA	4.18	3.79	3.85	4.14	4.44	1.066	0.823	0.528	0.329
TP	59.80	55.67	49.05	55.07	53.82	7.666	0.225	0.216	0.129
TG	0.22	0.22	0.18	0.21	0.20	0.055	0.642	0.377	0.478
TC	2.60	2.25	2.16	2.44	2.45	0.452	0.487	0.839	0.132
LDLC	0.60	0.53	0.45	0.54	0.57	0.163	0.596	0.845	0.155
LDLC	0.76	0.64	0.64	0.72	0.68	0.133	0.453	0.659	0.248
GLU	3.45	3.36	2.88	3.50	3.32	0.759	0.634	0.912	0.43

TP ^1^: Total protein; TG ^2^: Triglycerides; TC ^3^: Total cholesterol; LDLC ^4^: Low-density lipoprotein cholesterol; HDLC ^5^: High-density lipoprotein cholesterol; GLU ^6^: Blood glucose. ^a^^, b^ Means within a row with no common superscripts are significantly different (*p* < 0.05).

**Table 6 animals-12-01884-t006:** Effect of dietary SID lysine levels on the concentration of free amino acids in serum of Baqing pigs at 20–90 kg (mmol/L).

Item	Treatment					SEM	*p*-Value		
	T1	T2	T3	T4	T5		GLM	Linear	Quadratic
Lysine	0.14 ^c^	0.27 ^ab^	0.31 ^a^	0.20 ^bc^	0.25 ^ab^	0.07	0.003	0.125	0.010
Aspartate	0.031	0.033	0.042	0.033	0.032	0.008	0.220	0.804	0.087
Threonine	0.21	0.21	0.21	0.16	0.16	0.066	0.491	0.109	0.664
Serine	0.14	0.16	0.19	0.11	0.15	0.043	0.070	0.635	0.346
Glutamate	0.27	0.26	0.27	0.21	0.25	0.063	0.437	0.273	0.792
Glycine	1.64	1.71	1.91	1.44	1.80	0.423	0.401	0.917	0.890
Alanine	0.56 ^b^	0.54 ^b^	0.71 ^a^	0.43 ^b^	0.53 ^b^	0.121	0.012	0.280	0.255
Valine	0.16	0.15	0.16	0.19	0.20	0.049	0.318	0.078	0.357
Cysteine	0.003	0.002	0.002	0.003	0.002	0.003	0.884	0.640	0.749
Methionine	0.03	0.03	0.04	0.02	0.02	0.013	0.103	0.071	0.088
Isoleucine	0.09	0.07	0.09	0.08	0.09	0.019	0.380	0.446	0.560
Leucine	0.21	0.19	0.23	0.19	0.23	0.032	0.074	0.526	0.200
Tyrosine	0.07	0.07	0.07	0.06	0.06	0.014	0.128	0.028	0.807
Phenylalanine	0.09	0.09	0.11	0.09	0.09	0.015	0.084	0.600	0.111
Tryptophan	0.01	0.02	0.01	0.01	0.01	0.016	0.987	0.906	0.717
Histidine	0.04 ^b^	0.03 ^b^	0.04 ^b^	0.05 ^ab^	0.08 ^a^	0.023	0.015	0.002	0.085
Arginine	0.18	0.18	0.20	0.16	0.15	0.036	0.175	0.161	0.132
Proline	0.42	0.43	0.48	0.31	0.45	0.108	0.117	0.732	0.923

^a^^, b^^, c^ Means within a row with no common superscripts are significantly different *(**p* < 0.05).

**Table 7 animals-12-01884-t007:** Effect of dietary SID lysine levels on carcass characteristics of Baqing pigs.

Item	Treatment		SEM	*p*-Value
	T3	T4		
Final body weight, kg	88.75	92.22	1.67	0.093
Dressing percentage, %	73.245	73.038	0.845	0.872
Carcass weight, kg	65.008	67.325	0.963	0.200
Carcass length, cm	84.167	86.167	0.832	0.121
Lean meat percentage, %	40.413	41.218	0.874	0.540
10th-rib backfat thickness, cm	3.835	3.878	0.118	0.800
Loin area, cm^2^	40.412	41.715	3.0260	0.769

**Table 8 animals-12-01884-t008:** Effect of dietary SID lysine levels on meat traits of Baqing pigs.

Item	Treatment		SEM	*p*-Value
	T3	T4		
Drip loss, %	1.233	1.375	0.180	0.594
Cooking loss, %	35.845	35.175	0.609	0.455
Shear force, kg	3.707	4.517	0.169	0.007
Marbling score	2.333	2.333	0.211	1.000
Intramuscular fat, %	3.007	3.487	0.241	0.209
pH_45min_	6.487	6.412	0.113	0.660
pH_24h_	5.775	5.780	0.066	0.959
Color parameters	3.833	3.667	0.189	0.549
Lightness (L*)	31.592	31.232	0.465	0.599
Redness (a*)	9.830	9.703	0.330	0.792
Yellowness (b*)	5.597	5.887	0.204	0.973
Taste score				
Meat flavor	2.917	2.862	0.165	0.753
Meat tenderness	2.947	2.777	0.281	0.572
Meat juiciness	2.582	2.430	0.254	0.576
Meat cooked degree	4.082	3.943	0.1003	0.226

## Data Availability

The datasets analyzed in the current study are available from the corresponding author on reasonable request.

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
