# Peer review of "Effects of Dietary Lysine Levels on Growth Performance, Nutrient Digestibility, Serum Metabolites, and Meat Quality of Baqing Pigs"

_animals, 2022, doi:10.3390/ani12151884_

Round 1

Reviewer 1 Report

Dear authors,

Thank you very much for the interesting article. Mine concerns are minor and they are indicated within the text.

The paper presents novel and valuable findings. The introduction provides evidence-based background for the research. The methods have been adequately described, results are well presented and data interpretation is appropriate. The findings are thoroughly discussed, and conclusions are justified by the results. I did not find any factual errors. 

All the best and stay safe,

Reviewer 2 Report

Dear editor

Pls see the attached file.

REGARDS

Reviewer 3 Report

Manuscript animals-1735896, entitled “Effects of Dietary Lysine Levels on Growth Performance, Nutrient Digestibility, Serum Metabolites, and Meat Quality of Baqing Pigs”

Recommendation:       The above paper is not suitable for publication in its present form.

General comments:

1)      The article provides information about the effects of dietary lysine levels on growth performance, nutrient digestibility, serum metabolites, and meat quality of Baqing pigs. However, there are a lot of grammar, stylistic and syntax errors. Language needs improvement. Please check L53-54, 68-69, 184-186, 256-259, 276-277, 314-318, 334-335

2)      Please include details concerning the equipment and methods used for meat quality assessment (pHmeter, chromameter, dynamometer etc). What about digestibility methods?

3)      Why were only T3 and T4 groups selected for carcass and meat quality assessment? Since only two groups were used, it is not appropriate to state that “Increasing dietary Lys content increased the shear force of the longissimus dorsi muscle (P < 0.05), whereas it did not affect the other carcass and meat traits.” (L42-44)

4)      When P-value is >0.05, no superscripts are used. Please correct Table  3 “Phase 1 – ADFI, F/G”, “Phase 3 – ADFI”, Table 4 “Phase 2 – GE, CA”, Table 5 “Phase 1 – TP, HDLC” “Phase 2 – LDLC”, Table 6 “Leucine, Phenylalanine”

5)      You reached to the conclusion that “The optimal SID-Lys requirement of 20~40 kg, 40~60 kg, and 60~90 kg of Baqing pigs fed corn-soybean meal-based diets is estimated to be 0.92 %, 0.66%, and 0.55% of the diets, respectively.” (L44-46). Please explain in detail how.

6)      Please check reference style throughout the text. The year in parenthesis is not necessary, only the number in [ ]. For example, in L70 “Andretta et al. [9] stated that dietary Lys supplementation can maintain the dietary amino acid balance, increase feed conversion rate and indirectly reduce nitrogen excretion.”

Specific comments:

L16: “…of different standardized ileal digestible (SID) lysine…”

L21-22: This is not completely correct. Please check the 3rd general comment

L25-26: “This study was carried out to determine the lysine requirements of Baqing pigs and the effects of different dietary lysine levels on growth performance…”

L27: “selected” instead of “blocked”

L28: “…assigned to five dietary treatments, each with six replicates…” I think that authors mistakenly written 5 replicates, because 5 * 4 * 5 = 100 and not 120 and in L100 state that 6 replicates existed.

L27-28 and throughout the text: The pigs were castrated or half female? What do you mean?

L35: “The digestible energy (DE) was increased….”

L35: Also in second phase? Are you sure? Please check Table 4.

L37-38: “and quadratically increased the serum GLU concentration” Increased or decreased?

L38, 42 and throughout the text: “linearly” instead of ‘linear” and “quadratically” instead of “quadratic”

L39: Please delete “And the”

L41: Only in T3

L52: “bred” instead of “cultivated”

L53: “due to” instead of “because of” and “robustness” instead of “adaptation”

L56: “that limits” instead of “and hinders”

L57-58: “Currently, diets for Baqing pig are mainly formulated based on the nutrition recommendation tables of National…”

L59: Please delete “data”

L60-62: “Obviously, it is not appropriate due to the different characteristics of this indigenous breed [5,6]. Therefore, it is necessary to find the optimal nutritional requirements of Baqing pig for the improvement of its production performance and the promotion of indigenous…”

L67-68: Please delete “(National Research Council [NRC], 2012)”

L70: “stated” instead of “elucidated”

L71: Increase FCR? Are you sure? Is this a positive effect? Do you mean “decrease”?

L79: “In growing pigs, deficiency or excess of dietary Lys cannot only…”

L88: “…and fattening to provide scientific…”

L98: “…with an age…”

L99: “…randomly to five…”

L118: Please delete “Controls and Sampleing”

L120: “…after overnight fasting.”

L122: G/F or F/G or FCR?

L122, 133: “free access to water” instead of “drank freely”

L123: “…blood samples. 10 mL from each pig were collected by jugular…”

L125: “At the end of each phase, fecal…”

L127: “diet” instead of “meal”

L130: Six per treatment is too small sample size

L143: Please delete “(AOAC 2005)”

L159: Please delete “and the identity link function”

L177: “…period had the greatest numerical value in T4 group.”

L208: and TG? Please check Table 5 and L205

L209: Please delete “the”

L222: Only in T3

L223: I think that we cannot reach to safe conclusions

L239: Do you mean “The present study”?

L250: G/F or F/G or FCR?

L251: What do you mean by “growing period”? Phase 1 or 2?

L254: “effects on” instead of “modification of”

L269: What do you mean by “large-physique”? Large-sized?

L284: Momentous?

L297: In these studies or in the present study?

L308-309: In what phase?

L309: “T4” instead of “treatment four”

L323: “denotes” instead of “signifies”

L338: “…with increased dietary Lys levels,”

L340: “Morrison et al. (1961) stated that dietary Lys deficiency reduced…”

L342: Interrelated?

L350: “…in protein deficient diets could reduce…”

L354: A greater shear force value is desirable? How is it related with tenderness?

Round 2

Reviewer 3 Report

Authors made the majority of the necessary amendments. However, some points should be corrected before the acceptance of the article.

L30, 100: The term "half female" is not correct. Please use "castrated" alternatively

L54: What do you mean by "rough feeding resistance"?

L68-70: Please rephrase

L71: Increase or improve?

L154: Please add reference

L201-202: Please rephrase

L240: Please rephrase

L284: Please delete "(2012)"

L334-336: Please rephrase

L377-378: Please rephrase
